# Learning Sparse Representations of Preferences within Choquet Expected Utility Theory

**Margot Herin**[1]                **Patrice Perny**[2]                **Nataliya Sokolovska**[3]

[1,2]Sorbonne University, CNRS, LIP6, UMR 7606 , 4 place Jussieu, 75005 Paris, France
[3]Sorbonne University, INSERM, NutriOmics, UMR S 1269, 91, Boulevard de l'Hôpital, 75013 Paris, France

## Abstract

This paper deals with preference elicitation within Choquet Expected Utility (CEU) theory for decision making under uncertainty. We consider the Savage's framework with a finite set of states and assume that preferences of the Decision Maker over acts are observable. The CEU model involves two parameters that must be tuned to the value system of the decision maker: a set function (capacity) modeling weights attached to events, of size exponential in the number of states, and a utility function defined on the space of outcomes. Our aim is to learn a sparse representation of the CEU model from preference data. We propose and test a preference learning approach based on a spline representation of utilities and the sparse learning of capacities to obtain CEU models achieving a good tradeoff between the aim of sparsity and the expressivity required by preference data.

## 1 INTRODUCTION

Decision theory has developed an entire stream of theoretical works on the axiomatic foundations of preference models either for descriptive, normative or prescriptive purposes [von Neumann and Morgenstern, 1947, Savage, 1954, Fishburn, 1970, Gilboa, 2008, Quiggin, 2012, Wakker, 2013]. The mathematical models used to describe preferences include parameters that can be fitted to the value system of the decision maker (DM). The role of these preference parameters is well understood and the decision behavior of an individual can be interpreted by analysing the values of these parameters. For example, in expected utility theory, risk aversion is equivalent to the concavity of the utility function and the level of risk-aversion of an individual can be measured from the curvature and the slope of his/her utility function [Arrow, 1971, Pratt, 1978].

In the framework of decision under uncertainty (i.e., no probability of the events needs to be given) and risk (i.e., probabilities of the events are known) various models have been proposed, involving an increasing number of preferential parameters to cover an ever larger class of decision behaviors. For example, Tversky and Kahneman [1979] have observed a frequent violation of the *sure thing principle* of Savage [1954] in their experiments on preferences. Such violations preclude any representation of the observed preferences by Expected Utility (EU). Then, rank-dependent models have been introduced relying on a weakened version of the sure thing principle. Among them, Choquet Expected Utility (CEU) [Schmeidler, 1989] has received much attention due to its high descriptive possibilities and the fact that it boils down to well known simpler models for some well identified subclasses of capacities or utilities. Among them let us mention rank-dependent utility (RDU) [Quiggin, 2012], Yaari's model [Yaari, 1987] and EU [Savage, 1954].

CEU can easily explain standard preference inversions as those observed in Allais paradox (in the context of risk) and in Ellsberg paradox (in the context of uncertainty) [Ellsberg, 1961, Wakker, 2001]. However, the CEU model requires the definition of a non-necessarily additive set function (named capacity) that assigns a weight to every event that may occur in the problem, in addition to the utility function. In a problem where uncertainty is represented by a finite set $S$ of $n$ states of nature, the set of events under consideration is the set of all subsets of $S$. Hence the definition of the capacity and therefore of CEU requires the determination, in the general case, of $2^n$ coefficients, in addition to those that are necessary for the definition of the utility function.

The growing number of preference parameters, justified by descriptive objectives, comes at a cost: sophisticated decision models are harder to learn and need larger bases of preference data to be able to make reliable predictions on new data. Since preference data are usually not very numerous in practical applications and may be costly to obtain (preference queries must be asked to the DM or derived from a history of previous decisions) there is a

*Accepted for the 38th Conference on Uncertainty in Artificial Intelligence* (UAI 2022).

need of flexible approaches allowing to adapt the number of preference parameters used in the model to the expressivity required by preference data. This question is of particular relevance for CEU due to its expressivity but also to its lack of compactness in the general case. For this reason, we study in this paper the potential of sparse learning to determine compact instances of CEU from small preference databases.

One possible source of difficulties here is the interplay of utilities and capacities in the computation of CEU values, making the learning of these two types of parameters interdependent. Another difficulty comes from the fact that utilities and capacities are not assumed to be directly observable and should be derived from preference statements over pairs of alternatives. Taking these specificities into account, we propose a learning approach that proceeds in two steps: using preference questions specially designed for utility elicitation we learn a spline representation of the utility function and then derive a sparse representation of the capacity from further preference examples.

The paper is organized as follows: Section 2 introduces the CEU model and some basic concepts and properties related to Choquet integrals. In Section 3 we review some works on learning decision models based on Choquet integrals and introduce the premises of a two-phase approach to learn utilities and then capacities from preference data. Then the learning of the utility function is presented in Section 4 and the learning of the capacity in Section 5. In the two latter sections we present numerical tests to show the effectiveness of the proposed approach.

## 2 BACKGROUND ON CEU

We adopt the standard setting of Savage [1954] for decision making under uncertainty. We have to compare acts the outcomes of which depend on the (unknown) state of nature. Here we consider a finite set of states $S = \{1, \ldots, n\}$ that is supposed to include all relevant possible futures. Any subset $A \subseteq S$ must be interpreted as an event. For instance, if $S = \{1, 2, 3\}$ the set $A = \{1, 3\}$ represents the event "$s = 1$ or $s = 3$" where $s$ is the actual state of nature.

The acts to be compared are seen as functions defined from $S$ to the outcome space $X$. For simplicity we assume here that outcomes are payoffs and that $X$ is the real line. Any possible act $x$ is characterized by an outcome vector $(x_1, \ldots, x_n)$ where $x_i$ is the outcome of $x$ in state $i$, for $i = 1, \ldots, n$. We will denote $\mathcal{X} = X \times \ldots \times X$ the homogeneous cartesian product containing all possible acts given $S$ and $X$. Within $\mathcal{X}$ we distinguish constant acts denoted $\bar{x} = (x, \ldots, x)$ for any $x \in X$ (their outcome does not depend on the state of nature). We also define a mixture of acts as follows: for any $A \subset S$ and for any two acts $x, y \in \mathcal{X}$,

let $xAy$ denote the act of $\mathcal{X}$ defined by:

$$(xAy)_i = \begin{cases} x_i & \text{if } i \in A \\ y_i & \text{otherwise} \end{cases} \quad i = 1, \ldots, n.$$

It represents an act whose possible outcomes are those of $x$ if event $A$ occurs and those of $y$ otherwise. Mixtures of constant acts of type $\bar{x}A\bar{y}$, $x, y \in X$ are binary acts the outcome of witch is $x$ if $A$ occurs and $y$ otherwise. They are useful to design informative preference queries in the elicitation of utilities as we will see later in the paper.

Now, we introduce the Choquet Expected Utility model in the context of Savage. It is defined from two parameters: a utility function $u$ modeling the sensitivity of the DM with respect to outcomes and the capacity $v$ which is a set function defined on $2^S$, monotonic w.r.t to set inclusion (i.e., $v(A) \le v(B)$ whenever $A \subseteq B \subseteq S$) and normalized (i.e., $v(\emptyset) = 0$ and $v(S) = 1$) modeling the sensitivity of the decision maker towards uncertainty (its chance attitude e.g., optimism or pessimism, see Wakker [2001]). Given these two parameters $u$ and $v$, the CEU model assigns to every act $x$ an overall value $f_v^u(x)$ defined as the discrete Choquet integral of the utility of the outcome vector which reads as follows:

$$\begin{aligned} f_v^u(x) &= \sum_{i=1}^n \big[v(X_{(i)}) - v(X_{(i+1)})\big]u(x_{(i)}) \quad (1) \\ &= \sum_{i=1}^n \big[u(x_{(i)}) - u(x_{(i-1)})\big]v(X_{(i)}) \quad (2) \end{aligned}$$

where (.) is any permutation of $S$ such that $x_{(1)} \le \ldots \le x_{(n)}$, and $X_{(i)} = \{(i), \ldots, (n)\}$ is the event "the outcome of $x$ is greater or equal to $x_{(i)}$" for $i = 1, \ldots, n$. Furthermore we assume that $x_{(0)} = 0$ and $X_{(n+1)} = \emptyset$. For example, if $S = \{1, 2, 3\}$ then $f_v^u(100, 10, 60) = u(10)v(\{1, 2, 3\}) + [u(60) - u(10)]v(\{1, 3\}) + [u(100) - u(60)]v(\{1\})$ by Equation 2.

CEU theory provides an axiomatic framework under which the DM's preferences $\succsim$ over acts are represented by $f$ [Schmeidler, 1989, Gilboa, 2008]. Formally we have: $x \succsim y$ iff $f_v^u(x) \ge f_v^u(y)$. Let us briefly recall some key properties that illustrate the role of the capacity in the model:

- the monotonicity of $v$ is required to make sure that $f_v^u(x) \ge f_v^u(y)$ when $x_i \ge y_i$ for all $i \in S$.
- $CEU$ boils down to Savage's expected utility when $v$ is additive (i.e., $v(A \cup B) + v(A \cap B) = v(A) + v(B)$ for all $A, B \subseteq S$).
- the preference induced by $f$ satisfies *uncertainty aversion* if and only if $u$ is concave and $v$ supermodular (i.e., $v(A \cup B) + v(A \cap B) \ge v(A) + v(B)$ for all $A, B \in S$) [Chateauneuf and Tallon, 2002]. Uncertainty aversion (a.k.a convexity of preferences) reads as follows: if the DM is indifferent between $x$ and $y$ (denoted $x \sim y$) then $\alpha x + (1 - \alpha)y$ will be preferred to $x$ (and also to $y$ by symmetry) for any $\alpha \in [0, 1]$. The convex mixture of $x$ and $y$ reduces the uncertainty of outcomes w.r.t $x$ and $y$ and makes the DM better off.

- *CEU* boils down to the rank-dependent utility for decision making under risk whenever $v(A) = w(p(A))$ where $p$ is the probability measure on events and $w$ is a monotonic weighting function such that $w(0) = 0$ and $w(1) = 1$ [Quiggin, 2012]. If in addition $u$ is linear then CEU boils down to Yaari's model [Yaari, 1987].

Another useful formulation of the CEU model relies on the Möbius inverse of the capacity. The Möbius inverse of $v$ is another set function $m$ defined on $S$ by: $m(A) = \sum_{B \subseteq A} (-1)^{|A \setminus B|} v(B)$ for all $A \subseteq S$. The coefficients $m(A)$ are called Möbius masses, they completely characterize $v$. We indeed have $v(A) = \sum_{B \subseteq A} m(B)$. The values of $m$ can be positive or negative but add up to 1 since $\sum_{B \subseteq S} m(B) = v(S) = 1$. When $v$ is additive the only non-null Möbius masses are those of singletons.

Interestingly enough, the CEU model can be directly expressed from the Möbius inverse [Chateauneuf and Jaffray, 1989] by:

$$f_v^u(x) = \sum_{B \subseteq S} m(B) \min_{i \in B} \{u(x_i)\} \qquad (3)$$

This formulation shows that $f_v^u(x)$ might admit a compact representation whenever the Möbius inverse is sparse. A frequent option used to handle capacities with a sparse representation is to require that Möbius masses vanish for all subsets of states larger than a given $k$ smaller than $n$. In this case, the resulting capacity is said to be $k$-additive [Grabisch, 1997] and admits a more compact representation than in the general case. For instance, when the capacity is 1-additive then all Möbius masses are null except for singletons where they are positive due to monotonicity. However, in this case, Equation 3 shows that $f$ boils down to an expected utility with a significant loss of expressivity.

A more interesting tradeoff could be obtained with $k$-additivity for some small value of $k$ larger than 1 but it seems difficult to select a suitable value of $k$ without looking at preference data. Moreover it may happen that very sparse but still $n$-additive capacities perfectly match preference data as illustrated in the following:

**Example 1.** *Assume that the DM is pessimistic and behaves according to the min criterion refined by an expectation to ensure strict monotonicity w.r.t Pareto dominance. This behavior can be described by $f(x) = (1 - \epsilon) \min\{u(x_i), i \in S\} + \epsilon \sum_{i=1}^{n} p_i x_i$ where $p_i$ are subjective (positive) probabilities and $\epsilon > 0$ is chosen arbitrarily small. $f$ is an instance of CEU obtained from Equation 3 with $m(\{i\}) = \epsilon p_i$, $\forall i = 1 \dots n$, $m(S) = 1 - \epsilon$ and $m(B) = 0$ for all $B \subset S$ such that $|B| > 1$.*

Example 1 shows that preferences induced by $f$ could not be properly described nor approximated by a $k$-additive capacity with $k < n$ (because of the drop of the most important term of weight $1 - \epsilon$). Yet, $f$ can be closely approximated

with the min criterion which admits a very sparse Möbius inverse representation. This calls for a more efficient approach to derive sparse representations of CEU from preference data. This question will be adressed later in the paper.

In order to illustrate both the descriptive potential of CEU and its ability to admit a sparse representation in terms of Möbius inverse we now consider a standard urn example due to Ellsberg [1961].

**Example 2.** *An urn contains 90 balls including 30 red, and 60 blue or yellow balls in unknown proportion. We consider four bets, on the one hand $x$ (resp. $y$) yielding 100 if the drawn ball is red (resp. blue), and on the other hand $z$ (resp. $w$) yielding 100 if the drawn ball is not blue (resp. not red). Here $S = \{R, B, Y\}$ for red, blue, yellow, and the acts under consideration are $x = (100, 0, 0)$, $y = (0, 100, 0)$, $z = (100, 0, 100)$ and $w = (0, 100, 100)$. Note that the pair $(x, y)$ compares similarly to the pair $(z, w)$ except that the common outcome attached to yellow balls moves from 0 to 100. Despite this similarity, most of people prefer $x$ to $y$ but $w$ to $z$. It can easily be checked that such preferences are not representable by EU.*

*However, these preferences can be represented by CEU. Let us assume that $u(0) = 0$ and $u(100) = 1$ and $v(\{R\}) = 1/3, v(\{B\}) = v(\{Y\}) = 0$, $v(\{R, B\}) = v(\{R, Y\}) = 1/3$, $v(\{B, Y\}) = 2/3$ and $v(\{R, B, Y\}) = 1$. Note that for all events, $v$ yields the lower possible probability of the event according to our knowledge of the urn content. We have $f_v^u(x) = 0v(\{R, B, Y\}) + (1-0)v(\{R\}) = 1/3$. Similarly we obtain $f_v^u(y) = 0$, $f_v^u(z) = 1/3$ and $f_v^u(w) = 2/3$. Hence, $f_v^u(x) > f_v^u(y)$ and $f_v^u(w) > f_v^u(z)$ which is consistent with the observed preferences. Moreover, the Möbius inverse of $v$ is everywhere 0 except that $m(\{R\}) = 1/3$ and $m(\{B, Y\}) = 2/3$. Hence we get a sparse representation of $f$ that fits the observed preferences: $f_v^u(x_1, x_2, x_3) = u(x_1)/3 + 2 \min\{u(x_2), u(x_3)\}/3$.*

We end the section by mentioning a third representation of the capacity based on interaction indices, $I(A)$ for all $A \subseteq S$ [Grabisch, 1997] that will be discussed later in the paper. When $A$ is a singleton $\{k\}$, the interaction index $I(\{k\})$ is nothing else but the so-called Shapley value measuring the average marginal increment $v(B \cup \{k\}) - v(B)$ taken on all events $B \subseteq S$ that do not contain $k$. The notion of Shapley interaction index extends to any subset $A$ of $S$. Interaction indices can be uniquely defined from $v$ or $m$ indifferently. Conversely $v$ and $m$ can be obtained from $I$. For more details see Grabisch [1997]. In the case of Example 2, the interaction indices are given by $I(\emptyset) = 7/18$, $I(\{1\}) = I(\{2\}) = I(\{3\}) = 1/3, I(\{2.3\}) = 2/3$, the other coefficients being null.

# 3 LEARNING THE CEU MODEL

## 3.1 RELATED WORK

Fitting the parameters of a decision model based on a Choquet integral to observed preferences is a question present both in the literature on decision theory (preference elicitation) and in the literature on machine learning (preference learning). In the context of decision making under risk, some elicitation protocols proposing a series of preference queries involving pairs of lotteries have been proposed to construct methodically a set of points on the utility curve and then on the probability weighting function defining the capacity in the RDU model and in cumulative prospect theory (CPT) [Wakker and Deneffe, 1996, Abdellaoui, 2000].

Another stream of work developed in the literature on multicriteria decision aid concerns the use of non-linear regression for the identification of the capacity from overall evaluations prescribed by the DM, and the use of ordinal regression method from preference examples, assuming the utilities are known [Grabisch et al., 2008, Grabisch and Labreuche, 2010]. The prior construction of utility in this setting is often based on direct queries on difference of attractiveness between attribute values, see e.g., the Macbeth method [Bana e Costa and Vansnick, 1997].

Another approach developed in AI consists in progressively reducing the uncertainty about the preference parameters. A first set of methods proceeds by successive reductions of the parameter space using preference queries adaptively selected for their information value (e.g., using the minimax regret criterion). This incremental approach was used for the identification of utilities in [Wang and Boutilier, 2003], for the identification of the probability weighting function [Hines and Larson, 2010, Perny et al., 2016], and for the identification of Choquet capacities in [Benabbou et al., 2017]. A second set of methods proposes another adaptive elicitation procedure based on a Bayesian approach used to iteratively revise a probability density on the parameter space, see e.g. [Chajewska et al., 2000, Bourdache et al., 2019, Gu et al., 2020].

None of the above mentioned contributions addresses the question of learning sparse representations of the capacity; however, some of them assume that capacities are $k$-additives for a prior reduction of model complexity.

The Choquet integral is also used in machine learning to replace the linear function of variables which is commonly used in standard regression methods [Gagolewski et al., 2019, Beliakov and Wu, 2020, Beliakov and Divakov, 2020]. For example, logistic regression was extended to Choquistic regression [Tehrani and Hülermeier, 2013, Tehrani et al., 2012a, 2011]. It is also used for learning to rank with the Choquet integral [Tehrani et al., 2012b] where the data are provided with the labels which are preference degrees from

an ordered categorical scale. The Choquet integral was also introduced as a kernel method [Tehrani, 2021].

In the machine learning community, statistical regularization is used to find a tradeoff between the model's generalizing performance and the model's complexity [Tibshirani, 1996, Hastie et al., 2015]. In particular, the Lasso method introduces the $L_1$ penalty to the objective function to obtain a sparse solution in a high dimensional setting with small number of observations. It can be used to obtain compact representations of capacities that include, in the general case, $2^n - 2$ free parameters. Several attempts have been made to reduce the complexity of the non-additive integrals via the $L_1$ penalty term. For example, the sparsity inducing penalty was applied to the capacity [Anderson et al., 2014, Adeyeba et al., 2015]; the penalised sum of squared errors with Gini-Simpson index regularisation and the $L_0$ norm on the Shapley values were considered in [Pinar et al., 2017]. The $L_1$ penalty was also applied to capacities represented by interaction indices in [de Oliveira et al., 2022].

The specificity of our approach is to learn from pairwise comparisons both a smooth utility function and a sparse Möbius representation of the capacity with no prior reduction of the class of admissible capacities, in the framework of decision making under uncertainty and CEU theory.

## 3.2 APPROXIMATING PREFERENCES WITH CEU

It is assumed here that utilities and capacities cannot be directly requested from the decision maker who may have no idea of the model under consideration. Moreover, overall values $f_v^u(x)$ are not assumed to be observable. We want to derive preference parameters from observed choices or from preference statements obtained from the DM on some pairs of alternatives. Standard preference statements are of type "$x$ *is at least as good as* $y$" (denoted $x \succsim y$), or "$x$ *and $y$ are indifferent*" (denoted $x \sim y$). Within CEU theory the weak preference $x \succsim y$ (resp. the indifference $x \sim y$) is interpreted as $f_v^u(x) \geq f_v^u(y)$ (resp. $f_v^u(x) = f_v^u(y)$).

We remark that the above inequalities and equalities are linear in $v$ for any fixed $x, y$ and $u$ by definition of $f$ (see Equation 2). We assume here that such preference statements are available, either because a preference database is available or because the DM is able to answer on demand to some preference queries.

Given a set of indifference statements $\mathcal{I} = \{(x^i, y^i) \in \mathcal{X}^2 : x^i \sim y^i, i = 1, \ldots, q\}$ and/or a set of preference statements $\mathcal{P} = \{(x^i, y^i) \in \mathcal{X}^2 : x^i \succsim y^i, i = 1, \ldots, p\}$, we look for a utility function $u$ and a capacity $v$ that match with the observed preferences. Since the CEU model may not perfectly match with the preferences expressed by the DM, we look for an approximate representation of preference data. The approximation problem can be formulated as follows:

$$\min \sum_{i=1}^{q} (\epsilon_i^+ + \epsilon_i^-) + \sum_{i=1}^{p} \epsilon_i \qquad (4)$$

$$\begin{cases} f_v^u(x^i) - f_v^u(y^i) + \epsilon_i^+ - \epsilon_i^- = 0, \ i = 1...q \\ f_v^u(x^j) - f_v^u(y^j) + \epsilon_i \geq 0, \ \forall(x^j, y^j), \ j = 1...p \\ v(A) \leq v(A \cup \{i\}) \forall i \in S, \forall A \subseteq S \setminus \{i\} \end{cases}$$

$$\epsilon_i^+ \geq 0, \epsilon_i^- \geq 0, \epsilon_j \geq 0, i = 1...q, j = 1...p.$$

The third line in the system of constraints is here to enforce the monotonicity of $v$ with respect to set inclusion. We remark that the above optimization problem is not linear since Choquet values of type $f_v^u(x)$ appearing in constraints include products of variables defining the utilities $u(x_i)$ and the capacities values $v(X_{(i)})$ (see Equation 1).

In many contributions on preference learning methods based on the discrete Choquet integral, the utility function is assumed to be known and the focus is made on fitting the capacity. In this case, all constraints of the approximation problem formulated in Equation 4 are linear in $v$ and the capacity can be obtained using standard linear programming solvers. This suggests learning the utility function first and then the capacity.

On the other hand, some recent contributions propose to learn simultaneously the utility function and the capacity. Finding exact solutions simultaneously both for the utility function and the capacity is a difficult task, since the problem is not linear and the constraints are not convex. Some heuristics to solve this problem were proposed. A stochastic method was introduced by Angilella et al. [2004], and Goujon and Labreuche [2013] discussed a fixed-point method where the problem is split into two iterative linear tasks. Another heuristic based on a linear approximation of the product of the utility functions with Shapley values and interaction indices was considered by Galand and Mayag [2017]. An approach to find an exact solution for both utilities and capacities (in the context of the Choquistic regression) was proposed by Tehrani et al. [2014] where the utility function was represented as a linear combination of sigmoid functions. More recently, Bresson et al. [2020] developed a neural architecture to learn both utilities and the corresponding parameters of hierarchical Choquet integrals.

As mentioned earlier, Wakker and Deneffe [1996], Abdellaoui [2000] have shown that, by a careful selection of preference queries, the utility can be indirectly observed and acquired, regardless the capacity. We would like to use this specificity of the CEU model to learn the utility in a first stage. Then, determining the capacity becomes easier and we can focus on learning sparse representations of $v$ in a second stage. In this respect, we now discuss the relative interest of several standard representations of $v$.

## 3.3 VARIOUS REPRESENTATIONS OF CAPACITY

In Section 2 we have mentioned two alternative representations of a general capacity $v$: the Möbius inverse $m$, and the interaction indices $I$. Let us compare now their ability to provide compact representations. First of all, we can remark that, if $v(\{i\}) > 0$ for some $i$, then $v(A) > 0$ for any event $A \supset \{i\}$. Moreover, $m$ is at least as compact as $v$ due to the following proposition.

**Proposition 1.** *Let $v$ be a capacity and $m$ its Möbius inverse, we have: $\|v\|_0 \geq \|m\|_0$, where $\|.\|_0$ denotes the $L_0$ norm, i.e., the number of non-zero coefficients.*

*Proof.* Consider a capacity $v$ and its Möbius representation $m$. If $v(A) = 0$ for some $A \subseteq S$, then $v(B) = 0$ for all $B \subseteq A$. Hence $m(A) = \sum_{B \subseteq A} (-1)^{|A \setminus B|} v(B) = 0$. Then $\{A : v(A) = 0\} \subseteq \{A : m(A) = 0\}$ and $\|v\|_0 = 2^n - |\{A : \mu(A) = 0\}| \geq 2^n - |\{A : m(A) = 0\}| = \|m\|_0$. $\square$

Moreover, the following result shows that the representation of $v$ in terms of interaction $I$ may lack of compactness e.g., when $v$ is a belief function (i.e., when $m$ is non-negative).

**Proposition 2.** *Let $v$ be a capacity and $m$ and $I$ its Möbius and interaction representations respectively. If $m$ is non-negative, then $\|I\|_0 \geq 2^{|T^*|}$, where $T^* = \text{argmax}_{T \subseteq S} \{|T| : m(T) > 0\}$.*

*Proof.* The interaction index $I$ is linked to $m$ by the following equation: $I(A) = \sum_{T \supseteq A} \frac{1}{t-a+1} m(T)$ for all $A \subseteq S$ [Grabisch, 1997]. Hence, for any $T$ s.t. $m(T) > 0$, we have $I(A) > \frac{1}{t-a+1} m(T) > 0$ for all $A \subseteq T$. Let $T^*$ be the subset of maximal cardinality among those such that $m(T) > 0$, then the $2^{|T^*|}$ subsets of $T^*$ have a strictly positive interaction index. Hence, $\|I\|_0 \geq 2^{|T^*|}$. $\square$

As an illustration, let us consider again the maximin criterion $f(x) = \min_i u(x_i)$ which is an instance of CEU obtained from Equation 3 with $m(A) = 0$ for all $A \subset S$ and $m(S) = 1$. Then the above proposition shows that $I(A) > 0$ for all $A \subseteq S$. In this case the $I$ representation is of size $2^n$ whereas the Möbius representation is very sparse (it includes a single non-null coefficient). Considering the above propositions and the well-known interest of Möbius masses to identify focal elements in beliefs, we focus hereafter on regularizations based on the Möbius representation of $v$ aiming to minimize $\|m\|_0$.

# 4 LEARNING THE UTILITY FUNCTION

## 4.1 ASSESSING UTILITIES FROM INDIFFERENCE STATEMENTS

Let us remind that the DM is not assumed to be able to provide the overall value of an act (this would amount to

directly asking utilities values). This is a source of difficulty because constraints of type $f_v^u(x) = \alpha$ frequently used to perform regressions cannot be obtained by questioning the DM. However the DM can be asked to compare any act $x$ to a constant act $\bar{y} = (y, \ldots, y)$ for some $y \in X$. Hence $x \succsim \bar{y}$ is equivalent to $f_v^u(x) \geq f_v^u(\bar{y}) = u(y)$. Whenever the DM is indifferent between $x$ and $\bar{y}$ we have $f_v^u(x) = u(y)$. In such a case, outcome $y$ is said to be the *certainty equivalent* of $x$. Indifference statements giving the certainty equivalent $y$ of binary acts of type $\bar{x}A\bar{z}$ for some $x, z \in X$ such that $x > z$ are often considered for preference elicitation. Indeed, $\bar{x}A\bar{z} \sim \bar{y}$ means that $f_v^u(\bar{x}A\bar{z}) = v(A)u(x) + (1 - v(A))u(z) = u(y)$. Hence, if $v(A)$ is known for some $A$, this enables to derive $u(y)$ from $u(x)$ and $u(z)$. Thus, $u$ might be constructed point by point on a given interval $[x_m, x_M]$ from such indifferences, starting with two reference values $u(x_m) < u(x_M)$ arbitrary selected (e.g., $u(x_m) = 0$ and $u(x_M) = 1$). This process was used to elicit utilities in the context of risk [Hines and Larson, 2010, Perny et al., 2016]. However, in our context we have no simple way to obtain $v(A)$ for some $A$ before knowing the utility function. For this reason we propose to learn the utility function by regression from indifference statements obtained with the tradeoff method [Wakker and Deneffe, 1996, Abdellaoui, 2000] adapted to the context of uncertainty.

The tradeoff method initially introduced in the context of risk involves preference queries using gambles. Here, we describe a counterpart of this method in the context of uncertainty, to assess the utilities of outcomes within a given interval $[x_m, x_M]$ using mixtures of constant acts. This method requires that there exists an event $A$ such that $\bar{x}_m \prec \bar{x}_m A \bar{x}_M \prec \bar{x}_M$. Within CEU theory these strict preferences translate into $f_v^u(\bar{x}_m) < f_v^u(\bar{x}_m A \bar{x}_M) < f_v^u(\bar{x}_M)$ which is equivalent to $0 = u(x_m) = f_v^u(\bar{x}_m) < u(x_m)(1 - v(\bar{A})) + v(\bar{A}) < f_v^u(\bar{x}_M) = u(x_M) = 1$, i.e., $0 < v(\bar{A}) < 1$.

So, given such an event $A$, let us choose $z, r, R \in X$ such that $z \leq x_M < r < R$ and consider the two following preference queries:

$Q(y, z)$: what is the outcome $y$ such that: $\bar{y}A\bar{R} \sim \bar{z}A\bar{r}$?
$Q(x, y)$: what is the outcome $x$ such that: $\bar{x}A\bar{R} \sim \bar{y}A\bar{r}$?

With such indifferences, the DM makes a tradeoff between upgrading $r$ in $R$ and downgrading $z$ in $y$ (or $y$ in $x$). Since $y \leq z$ and $x \leq y$, we have $f_v^u(\bar{y}A\bar{R}) = u(y)(1 - v(\bar{A})) + u(R)v(\bar{A})$ and $f_v^u(\bar{z}A\bar{r}) = u(z)(1 - v(\bar{A})) + u(r)v(\bar{A})$. Hence $f_v^u(\bar{y}A\bar{R}) = f_v^u(\bar{z}A\bar{r})$ implies $u(y)(1 - v(\bar{A})) + u(R)v(\bar{A}) = u(z)(1 - v(\bar{A})) + u(r)v(\bar{A})$ and therefore $(1 - v(\bar{A}))[u(z) - u(y)] = v(\bar{A})[u(R) - u(r)]$. Similarly, $\bar{x}A\bar{R} \sim \bar{y}A\bar{r}$ implies $(1 - v(\bar{A}))[u(y) - u(x)] = v(\bar{A})(u(R) - u(r))$. Since $v(\bar{A}) > 0$, if $u(R) - u(r) > 0$ then $u(z) > u(y) > u(x)$ and therefore $z > y > x$. Finally, we have $(1 - v(\bar{A}))[u(z) - u(y)] = (1 - v(\bar{A}))[u(y) - u(x)]$.

Since $v(\bar{A}) < 1$ we obtain:

$$u(z) - u(y) = u(y) - u(x) \qquad (5)$$

Such queries are often involved in a *standard sequence* that consists in sequentially asking question $Q(x_{i+1}, x_i)$ for $i = 0$ to $N - 1$, starting from $x_0 = x_M$ until $x_N \geq x_m$. From the observed indifferences, Equation 5 yields $u(x_{i-1}) - u(x_i) = u(x_i) - u(x_{i+1})$ and therefore $u(x_{i+1}) = 2u(x_i) - u(x_{i-1})$. Hence, fixing arbitrarily the utilities of $x_0$ and $x_1$ completely determines the utilities $u(x_i)$ for $i > 1$. However, if the DM makes some errors in assessing $x_i$ in the early steps of the sequence, these errors will propagate and impact the whole sequence [Blavatskyy, 2006]. In order to reduce the error propagation, we propose to perform a regression from a database of indifference statements obtained from queries of type $Q(y, z)$ and $Q(x, y)$ rather than performing a standard sequence.

More precisely, our learning approach proceeds as follows: for various non-null events $A$, various steps $s = R - r$ defined by different pairs $(r, R)$ and various $z$, the two preference queries $Q(y, z)$ and then $Q(x, y)$ are asked to the DM. The resulting database of indifference statements yields a set of necessary linear constraints on utility values given by Equation 5, for all triplets $(x, y, z)$ obtained from the answers to $Q$ queries. Then a regression by a monotonic spline is performed to identify the utility function that best fits the set of linear constraints on utility values.

## 4.2 MONOTONIC SPLINE REGRESSION UNDER UTILITY CONSTRAINTS

In order to represent the utility function, we use a monotonic spline function, i.e., a piecewise polynomial function of class $C^k$. Spline functions are widely used for data interpolation or approximation due to their ability to smoothly approximate complex shapes. Moreover they allow for a compact representation of utilities. Indeed, a spline function can be expressed as a linear combination of basis functions and is thus characterized by the coefficients of the combination. Since utility increases with payoffs, we will use a basis $(I_l)_{l=1}^L$ of monotonically increasing spline functions, known as I-spline functions [Ramsay, 1988] weighted by positive coefficients (adding up to 1 so as to have $u(x_M) = 1$). We use here cubic I-splines ($k = 3$) because they have matching first and second derivatives while preserving a local influence of every components. Formally, $u$ is defined by:

$$\forall x \in [x_m, x_M], \ u_\alpha(x) = \sum_{l=1}^L \alpha_l I_l(x) \qquad (6)$$

where $\alpha = (\alpha_1, \ldots, \alpha_L) \in [0, 1]^L$.

For the sake of illustration, we represent in Figure 1 the I-spline basis for $L = 10$ (value used in our tests) and an instance of utility function generated from this basis.

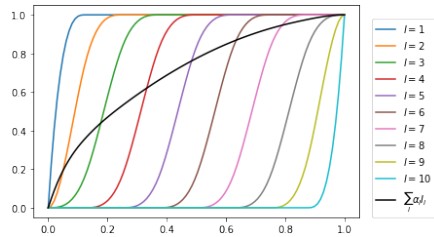

Figure 1: $u_\alpha(x)$ generated from the I-splines

Our observations have been obtained using the $Q$ queries leading to a database $\mathcal{B}$ of $N$ triplets $(x^i, y^i, z^i)$, as described in the previous subsection. We want to determine the parameters $\alpha_l$ that best fit the associated constraints $2u(y^i) - u(z^i) - u(x^i) = 0, i = 1, \ldots, N$ obtained from Equation 5. Hence, using Equation 6, the problem can be formalized as a linear program $P(\mathcal{B})$ with relaxed constraints ($N + 1$ constraints and $L + 2N$ variables):

$$P(\mathcal{B}): \min z = \sum_{i=1}^{N}(\epsilon_i^+ + \epsilon_i^-)$$

$$\begin{cases} \sum_{l=1}^{L} \alpha_l(2I_l(y^i) - I_l(z^i) - I_l(x^i)) + \epsilon_i^+ - \epsilon_i^- = 0, \forall i \\ \sum_{l=1}^{L} \alpha_l = 1 \end{cases}$$

$$\epsilon_i^+ \geq 0, \epsilon_i^- \geq 0, \alpha_l \geq 0.$$

Hereafter let $\alpha_l^*$ denote the optimal solution and $z^*$ the optimal value. Taking into consideration that the number of observations is always limited we need to assess the level of uncertainty of the utility function. To this end, we investigate a neighborhood of the optimal solution defined by $z^* \leq z \leq z^* + \delta$ where $\delta$ is a tolerance threshold. This neighborhood $V_\delta(z^*)$ contains all spline functions that satisfy the constraints on utilities with an error at most equal to $z^* + \delta$. The range of variation of utilities within this set is a good indicator of the level of uncertainty allowed by the constraints. It can be measured by the quantity $\rho = \max_{x \in [x_m, x_M]}\{\max_{\alpha \in V_\delta(z^*)} u_\alpha(x) - \min_{\alpha \in V_\delta(z^*)} u_\alpha(x)\}$ estimated by discretization of $[x_m, x_M]$. When $\rho$ is too large, the constraints are considered too weak to allow for the identifiability of $u$; one should carry on the $Q$ queries process. The elicitation procedure is formalized in Algorithm 1.

---

**Algorithm 1:** Utility elicitation with Q-queries

---
$i \leftarrow 0, B \leftarrow \emptyset$
**repeat**
    Select $A^i, x^i, R^i, r^i$
    Ask queries $Q(x^i, y^i), Q(y^i, z^i)$
    $\mathcal{B} \leftarrow \mathcal{B} \cup \{(x^i, y^i, z^i)\}$
    $(\alpha^*, z^*) \leftarrow P(\mathcal{B})$
    Compute $\rho$
    $i \leftarrow i + 1$
**until** $\rho \leq \epsilon$;

---

Let us illustrate our approach. We simulate a $Q$ queries process with a DM answering according to a given CEU model $f_v^u$. Answers to queries of type $Q(y, z)$ for a given $z$ and a pair $(r, R)$ are simulated by solving the equation $f_v^u(\bar{y}A\bar{R}) = f_v^u(\bar{z}A\bar{r})$ which gives $y = u^{-1}(u(z) + [u(R) - u(r)](v(A) - 1)/v(A))$. Then $x$ is derived from $y$ using a similar process to simulate the answer to question $Q(x, y)$. Then the resulting triplet $(x, y, z)$ is slightly distorted using a random uniform noise. This process is iterated $N$ times for randomly chosen $z, r, R$, and $A$. We used the mathematical programming solver Gurobi (version 9.1.2) to perform the optimization task. The result of the learning process is presented on Figure 2 where we increase the size of the database in order to reduce $\rho$. In this instance, $u$ is already well estimated with tight bounds for $N = 32$.

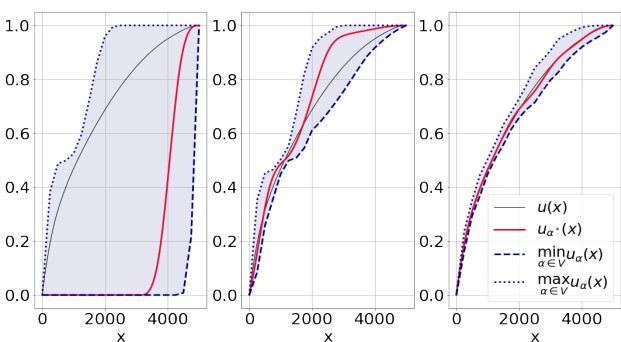

Figure 2: Identification of the utility function $u$ for $N = 4$, $N = 16$, $N = 32$ (left to right).

This experiment has been conducted for 1000 utility functions $u$ randomly generated in the space of spline functions. Below we show the decrease of $\rho$ and of the distance $d(u, u_{\alpha^*})$ between the estimated utility function $u_{\alpha^*}$ and $u$ as the number $N$ of learning examples increases. The distance is computed as the average absolute difference between both functions on a discretization of $[x_m, x_M]$.

| | N = 4 | N = 16 | N = 32 |
|---|---|---|---|
| $\rho$ | 0.687 | 0.124 | 0.072 |
| $d(u, u_{\alpha^*})$ | 0.354 | 0.024 | 0.004 |

Table 1: $\rho$ and $d(u, u_{\alpha^*})$ w.r.t the number of constraints $N$.

## 5 LEARNING THE CAPACITY

Given the utility function $u$ obtained as described in Section 4, we want to learn a sparse Möbius representation of the CEU model, based on Equation 3. However, since $\|m\|_1 \geq \sum_{B \subseteq S} m(B) = 1$ by definition, it is not quite natural to penalize with $\|m\|_1$ since its impossibility to decrease to zero would make ineffective any further reinforcement of the penalization as soon as $\|m\|_1 = 1$.

To overcome the problem we use the following representation of $m$: $m(B) = 1/n + w_B$ if $|B| = 1$ and $m(B) = w_B$ if $|B| > 1$, where $w_B$ are real coefficients (positive or negative) such that $\sum_{B \subseteq S} w_B = 0$ (hence $\sum_{B \subseteq S} m(B) = 1$). Note that when all coefficients $w_B$ are null, CEU boils down to a simple instance of Expected Utility where states are equally weighted. In the general case, $w$ represents the gap to this basic model. In order to obtain a sparse representation in terms of Möbius we penalise on $\|w\|_1$ instead of $\|m\|_1$. This leads to solve the following linear program, which is a regularized version of (4) reformulated with Möbius masses ($m_B$ are variables representing the masses $m(B), B \subseteq S$):

$$\min \sum_{i=1}^q (\epsilon_i^+ + \epsilon_i^-) + \sum_{i=1}^p \epsilon_i + \lambda \sum_{B \subseteq S} (w_B^+ + w_B^-)$$

$$\begin{cases} \sum_{B \subseteq S} m_B (U_B^{x^i} - U_B^{y^i}) + \epsilon_i^+ - \epsilon_i^- = 0, \ i = 1...q \\ \sum_{B \subseteq S} m_B (U_B^{x^i} - U_B^{y^i}) + \epsilon_i \geq 0, \ i = 1...p \\ m_B = 1/n + w_B, \ \forall B \subseteq S : |B| = 1 \\ m_B = w_B, \ \forall B \subseteq S : |B| > 1 \\ w_B = w_B^+ - w_B^-, \ \forall B \subseteq S \\ \sum_{C \subseteq B} m_{C \cup \{i\}} \geq 0, \ \forall i \in S, \ \forall B \subseteq S \setminus \{i\} \end{cases}$$

$$\epsilon_i^+ \geq 0, \epsilon_i^- \geq 0, \epsilon_j \geq 0, w_B^+ \geq 0, , w_B^- \geq 0, m_B, w_B \in \mathbb{R}$$

where $U_B^{x^i} = \min_{j \in B} \{u(x_j^i)\}, \forall B \subseteq S, \forall i$.

The number of variables and constraints are respectively $2^{n+2} + 2q + p$ and $q + p + 2^{n+1} + n2^{n-1}$. In practice, the LP above remains tractable because the number of states $n$ under consideration is generally low (at most a dozen).

Now, we share the results of our numerical experiments to illustrate the learned sparse model with the linear program described above. Here also the results are obtained with the Gurobi optimizer. First, we investigate how the generalizing performance evolves with the sparsity of $m$. We generated preference data as follows. A utility function $u$ and a Möbius-sparse capacity $v$ are randomly generated and preferences compatible with $f_v^u$ are generated. Training data take the form of $N$ pairs $(x^i, y^i)$ of acts whose outcomes are randomly drawn from $[x_m, x_M]$. The preferences are stated from this pairs as follows: let $\tilde{f}_v^u(x)$ be a perturbation of $f_v^u(x)$ by uniform noise randomly drawn from a given interval $[-\sigma, \sigma]$, for any act $x$. If the difference $|\tilde{f}_v^u(x^i) - \tilde{f}_v^u(y^i)| \leq \sigma$, then $x^i$ and $y^i$ are considered as indifferent. If the difference is greater than $\sigma$, we conclude to preference. Pairs with Pareto dominance are discarded.

For the sake of illustration, we present the results of our approach on a toy dataset with $n = 7$ states and $N = 100$ preference examples. Hence, we have $2^7 - 2$ parameters to learn. The learning is performed for various values of $\lambda$ (the weight of the regularization term) ranging from 0 to 100 in order to obtain a sequence of increasingly sparse representations. For each obtained model, we assess the performance in generalization by measuring the error rate on a test set

of 1000 preferences. Figure 3 (left) shows various possible tradeoffs between the test error and the compactness of the Möbius representation measured by $\|m\|_0$. The curve shows that the introduction of the penalization term relevantly reduces both the error in tests and the number of non-null masses up to a point where we get close to the true model. Beyond this point, we see that further enforcing sparsity is counterproductive and increases the error in test. Figure 3 (right) represents three Möbius representations of the capacity respectively learned without regularization ($\lambda = 0$, plot (1)), with regularization and optimal tradeoff ($\lambda = 0.5$, plot (2)), and the true one (plot (3)). It shows that the penalty term is needed to recover a model close to the true one.

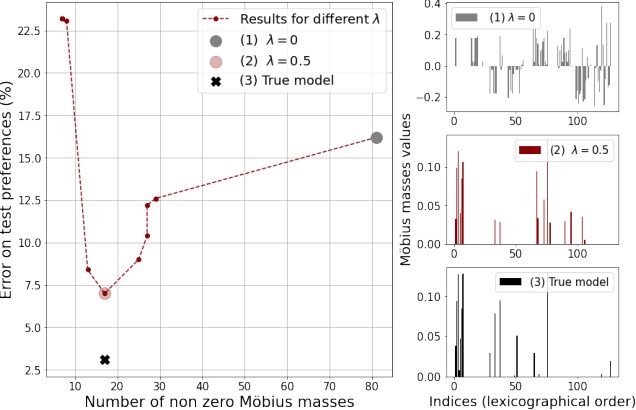

Figure 3: Test error versus $\|m\|_0$

A second experiment aims at highlighting the very special benefit of the regularization for small preferences databases in term of predictive performance. Figure 4 illustrates the advantage of sparse models (obtained by regularization) for settings where the number of preference examples is small. We observed the average error rate on datasets of increasing sizes ranging from $N = 50$ to $N = 1000$. The average is taken on 30 random datasets each time. We observe on Figure 4 that the smaller the dataset, the bigger the increase of performance obtained by regularized models.

Finally, we provide the result of a comparison between our approach (sparse regression) and a method based on 2-additive models (2-ADD) in Tables 2 and 3. We simulated 10 random models $f_v^u$ of dimension $n = 5$ and $n = 10$ and associated training sets of size $N = 70$ and $N = 400$ (and test sets of size 1000). We observe that our approach which adapts sparsity to preference data has significantly lower error rates than the method that enforces sparsity with 2-additivity. However, this advantage comes at an additional computational cost due to the increase of variables.

## 6  CONCLUSION

We have presented a new approach for learning the utility function and the capacity in CEU. A spline representation of

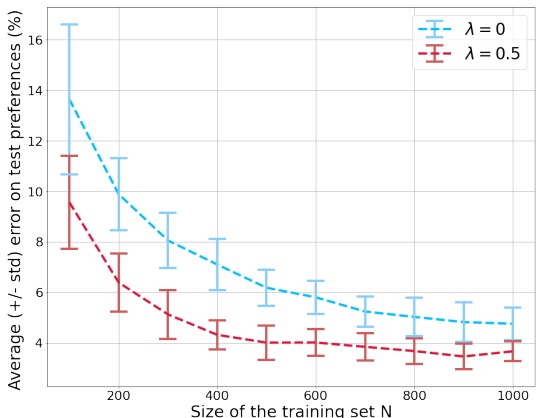

Figure 4: Comparative test error for sparse ($\lambda = 0.5$) and dense ($\lambda = 0$) models w.r.t the training set size $N$.

| $n$ | 5 | 10 |
|---|---|---|
| Sparse reg. | $5.98 \pm 4.36\%$ | $10.28 \pm 3.24\%$ |
| 2-ADD | $12.09 \pm 4.98\%$ | $15.98 \pm 1.18\%$ |

Table 2: Comparative average test error w.r.t $n$.

| $n$ | 5 | 10 |
|---|---|---|
| Sparse reg. | $0.0077 \pm 0.0021$ | $23.52 \pm 6.21$ |
| 2-ADD | $0.0066 \pm 0.0022$ | $0.35 \pm 0.06$ |

Table 3: Comparative average training time (sec) w.r.t $n$.

utilities is obtained via a regression from selected indifference learning examples. Then, a sparse representation of the capacity is obtained based on Möbius masses. Our tests confirm the practical effectiveness of the method. By proposing various tradeoffs between compactness and performance in the test phase, our approach allows the simplification of the general CEU model while maintaining the level of expressiveness required to describe the preference data.

A natural extension of this work is to develop an active learning version of our approach where the elicitation burden is oriented towards the determination of the best choice within a given set of alternatives. Also, an active selection of preference queries could reduce the number of examples required to learn a sparse representation of the capacity. Besides, we could extend our approach to the framework of multiattribute decision making. $Q$ queries could be adapted to learn a utility function per attribute using spline regression; then a sparse representation of the capacity could be learned to reveal the non-essential attributes and determine a multiattribute utility model keeping non-additive utilities only when necessary.

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
