# OpenReview forum: "Learning Sparse Representations of Preferences within Choquet Expected Utility Theory"
_auai.org/UAI/2022/Conference — UAI 2022 Poster_

### Official Review · Reviewer_vKfe · 2022-04-08

**Q2(1) Originality/Novelty:** 3
**Q2(2) Significance/Impact:** 2
**Q2(3) Correctness/Technical Quality:** 3
**Q2(6) Clarity Of Writing:** 3
**Q6 Overall Score:** 5
**Q8 Confidence In Your Score:** 4

**Q1 Summary And Contributions:**

This paper presents an approach for preference learning using Choquet expected utility (CEU) as the underlying descriptive model. The approach generally consists of a phase of learning utilities through an I-spline representation via a regression from indifference learning examples, followed by learning a sparse representation of “capacities” - this seems to rely on what the authors refer to as a Mobius mass formulation of CEU.

**Q2 Assessment Of The Paper:**

More detailed information regarding each of these aspects is given below:

**Q2(4) Quality Of Experiments (Optional):**

2: Fair: The experimental evaluation is weak: important baselines are missing, or the results do not adequately support the main claims.

**Q2(5) Reproducibility:**

2: Fair: Key resources (e.g., proofs, code, data) are unavailable but key details (e.g., proof sketches, experimental setup) are sufficiently well-described for an expert to confidently reproduce the main results.

**Q3 Main Strengths:**

There are a few things I liked about this paper:

1)	Often, papers of this sort do not make a distinction between prescriptive and descriptive decision making, but this one clearly states its motivation as being descriptive; specifically, the work is about making a case for the generality of a CEU as a descriptive theory, and then providing an approach for utility & capacity learning.
2)	The first half of the paper is well written, particularly Sections 2 and 3. I felt the authors covered a lot of ground and also explained many of the complicated ideas fairly clearly.
3)	The general problem of joint learning of utilities and capacities is a difficult one and relying on sparse representations seems quite reasonable.


**Q4 Main Weakness:**

I feel the paper has the following weaknesses:

1)	The experimental section is quite weak in my opinion. I expected a separate section that wud focus on experiments, and a more detailed and clearer demonstration of the benefits of the approach, possibly using a comparison to some baselines. Although I am not an expert in CEU preference elicitation/learning, the authors have cited some work which could plausibly have been used as baselines.
2)	I found the organization of the latter part of the paper to be confusing. There are results in Section 5.1 that in my view should have come much earlier in the paper. The significance of these results is lost on me. I feel the second half of the paper could be significantly improved, in clarity and perhaps also in experimental content.


**Q5 Detailed Comments To The Authors:**

Detailed comments follow:

There are various concepts mentioned in Section 1 without enough clarity for a generic reader, such as decision under uncertainty vs. risk, the sure thing principle, capacity, etc. Some of these ideas become clear later, such as capacity. More relevant and classic citations in Section 1 would also be helpful.

I find the phrase “sensitivity of the DM to …” to be a bit confusing.

In various places in Section 1, “the” is used when it is not needed, e.g. “the expected utility”.

More explanation immediately after eq. 3 would be helpful. The max-min criterion example is a good one, maybe it could become a separate example before the current Example 1, which is also a good one.

On p3, “decision models” should be “decision model”.

The authors may wish to fix the reference style in various places on p4.

There are obviously many more potential citations to refer to preference elicitation and learning in Section 2. The authors may wish to look into the following paper as well as papers it cites, since it also considers fitting a utility function while accounting for uncertainty: GaSPing for utility, Gu et al, AAAI 2020.

The first line of Section 3.2 doesn’t make any sense. In decision theory, utilities and probabilities can come from anywhere in general. I strongly suggest editing this line.

The last paragraph of Section 3.2 is a partial repeat from earlier. Also the connection to the trade-off method and the content of Section 4 only becomes clearer later.

Section 4 is too informal for my taste in parts, e.g. the last paragraph of Section 4.1 and the second last paragraph on p6. Could the procedures be shown more formally, say as an algorithm?

The table on p7 needs a number and a caption.

As mentioned earlier, I’m not sure about the positioning of Section 5.1 in the paper.

There needs to be a separate experiments section. As it stands, I’m not sure that I understood the benefits of the proposed approach. Also, readers need to be reminded of some of the parameters mentioned on p8, like N and \lambda.

**Q7 Justification For Your Score:**

If the experimental section had been stronger and there had been stronger evidence for the benefits of the proposed learning approach, this would have been a much stronger paper and a clear accept for me. As it currently stands, I find it to be borderline.

**Q9 Complying With Reviewing Instructions:**

1: Yes.

---

### Official Review · Reviewer_FTzx · 2022-04-10

**Q2(1) Originality/Novelty:** 2
**Q2(2) Significance/Impact:** 2
**Q2(3) Correctness/Technical Quality:** 2
**Q2(6) Clarity Of Writing:** 3
**Q6 Overall Score:** 5
**Q8 Confidence In Your Score:** 3

**Q1 Summary And Contributions:**

The authors provide a technique for learning representations of preferences within the Choquet expected utility model. The authors outline their main idea and provide some examples/experiments confirming that it works appropriately.

**Q2 Assessment Of The Paper:**

More detailed information regarding each of these aspects is given below:

**Q2(4) Quality Of Experiments (Optional):**

1: Poor: The experimental evaluation is flawed or the results fail to adequately support the main claims.

**Q2(5) Reproducibility:**

2: Fair: Key resources (e.g., proofs, code, data) are unavailable but key details (e.g., proof sketches, experimental setup) are sufficiently well-described for an expert to confidently reproduce the main results.

**Q3 Main Strengths:**

The paper gives a very good account of previous work and of Choquet expected utility approach

The paper fits the topic of the conference perfectly.

**Q4 Main Weakness:**

I am missing evidence that the problem studied is truly useful. There does not seem to be an experiment that would work on some (semi) realistic data to solve a meaningful problem. Hence, it is really hard to convince oneself that the problem that is being solved really needs solving.

**Q5 Detailed Comments To The Authors:**

Minor comments:
p. 3, left "larger that" --> "larger than"
p. 5, left "the Monte Carlo METHOD was"
p. 5, right "preferences queries" --> "preference queries"

**Q7 Justification For Your Score:**

The paper fits the topic of the confernece very well, but is missing any convincing argument that the studied problem is really important (except that many people have worked on similar problems).

**Q9 Complying With Reviewing Instructions:**

1: Yes.

---

### Official Review · Reviewer_o81Y · 2022-04-11

**Q2(1) Originality/Novelty:** 3
**Q2(2) Significance/Impact:** 2
**Q2(3) Correctness/Technical Quality:** 3
**Q2(6) Clarity Of Writing:** 3
**Q6 Overall Score:** 6
**Q8 Confidence In Your Score:** 3

**Q1 Summary And Contributions:**

This paper proposes an approach to learn the parameters of a Choquet integral (in the context on a problem on decision making under uncertainty). The utility function is first elicited, then the capacity. The paper suggest that this approach is useful when the capacity can be compactly represented by a mass function - is "sparse"  (as far as I understand, when the number of focal elements is low)

**Q2 Assessment Of The Paper:**

More detailed information regarding each of these aspects is given below:

**Q2(4) Quality Of Experiments (Optional):**

2: Fair: The experimental evaluation is weak: important baselines are missing, or the results do not adequately support the main claims.

**Q2(5) Reproducibility:**

2: Fair: Key resources (e.g., proofs, code, data) are unavailable but key details (e.g., proof sketches, experimental setup) are sufficiently well-described for an expert to confidently reproduce the main results.

**Q3 Main Strengths:**

The paper is well written and sound; the topic is interesting and fits UAI's domains - it can lead to interesting discussions. The approach is original.  Further developments and experiments are necessary, but if they confirm the significance of the approach, the result is valuable  (I agree with the fact that in the real wold capacity measures in general and belief functions in particular may have a limited number of focal elements)

**Q4 Main Weakness:**

The complexity of the learning process is unclear for me - how many comparisons are necessary ? (how may questions to the decision maker ?  in the example, n=7 (which is low), and the text suggests that 2^7 parameters have to be learnt ...  I understand that the approach is works (in terms of error rate) well when the number of focal elements is low, but (i) what about the number of examples needed and (ii)  what about its temporal complexity ?

Experiments are presented, but they do non compare the approach proposed (neither in terms of CPU nor in terms of error rate or size of the training set) to any existing method, despite the fact that several are discussed in the previous sections.


**Q5 Detailed Comments To The Authors:**

The paper is clear but for small details, e.g. a clear definition of sparsity (= number of focal elements ?) . Could you also provide a reference for equation (3) ?



**Q7 Justification For Your Score:**

As stated previously,  the approach is promising, targets an interesting question and seems sound. Nevertheless, its significance has to be worked out - the paper misses a complexity analysis  and an experimental comparison with previous approaches

**Q9 Complying With Reviewing Instructions:**

1: Yes.

---

### Official Review · Reviewer_brcZ · 2022-04-12

**Q2(1) Originality/Novelty:** 2
**Q2(2) Significance/Impact:** 2
**Q2(3) Correctness/Technical Quality:** 2
**Q2(6) Clarity Of Writing:** 3
**Q6 Overall Score:** 6
**Q8 Confidence In Your Score:** 3

**Q1 Summary And Contributions:**

The paper considers the problem of preference learning within the Choquet utility theory. The main contribution is learning sparse representations of the utility and capacity of the Choquet expected utility model from a set of preference statements. Specifically, these representations consist of spline functions (for utility) and Mobius masses (for capacity).


**Q2 Assessment Of The Paper:**

More detailed information regarding each of these aspects is given below:

**Q2(4) Quality Of Experiments (Optional):**

2: Fair: The experimental evaluation is weak: important baselines are missing, or the results do not adequately support the main claims.

**Q2(5) Reproducibility:**

2: Fair: Key resources (e.g., proofs, code, data) are unavailable but key details (e.g., proof sketches, experimental setup) are sufficiently well-described for an expert to confidently reproduce the main results.

**Q3 Main Strengths:**

1. The quality of the presentation is overall fairly good. The paper is relatively well organised and therefore it is fairly easy to follow.

2. Learning a CEU model from limited preference statements can be difficult in practice and therefore adopting a sparse representation of the capacity and utility functions involved in the model appears to be novel and therefore may help to alleviate some of these challenges.


**Q4 Main Weakness:**

1. The empirical evaluation appears quite weak. The experiments are carried out using only synthetic datasets. It would perhaps be good to expand the evaluation to include more realistic datasets.

2. The presentation of the learning algorithms needs additional details, such as a discussion about the complexity of the linear programs involved.

3. The Mobius transform in the context of the Choquet integrals for preference representation has been around for some time (e.g., [Mayag et al, 2011]). Therefore, it is not very clear what is the contribution of this paper beyond previous work.

[Mayag et al, 2011] B. Mayag, M. Grabisch, C. Labreuche. A representation of preferences by the Choquet integral with respect to a 2-additive capacity. In Theory and Decision 71, 297-324, 2011.


**Q5 Detailed Comments To The Authors:**

Learning the parameters of the sparse representations boils down to solving two linear programs, one for the spline functions and one for the Mobius masses. However, the presentation of the algorithms is somewhat high-level. Perhaps it would be good to add a small running example to illustrate how these linear programs are constructed. This will help the reader to have a better understanding of the proposed scheme.


**Q7 Justification For Your Score:**

The empirical evaluation appears weak and the contribution is not clear as the proposed representations for utility and capacity of the CEU model have been introduced in previous work.

**Q9 Complying With Reviewing Instructions:**

1: Yes.

---

### Decision · Program_Chairs · 2022-05-15

**Decision:**

Accept (Poster)

**Comment:**

Meta Review: The paper presents a new means to learn sparse Choquet integral, using for instance classical L_1 penalization techniques. This research participates to a long trend of research aiming at learning preference functions in various settings (here, having only limited preferential information).

All reviewers agree that the paper is well-written and the contribution nicely presented. Two main critics were done by the reviewers:

* The first is the potential significance of the research, i.e., that it addresses an actual real-world problem. After discussion and rebuttals, this critic appears to be of less importance

* The second (with which I would concur) is that the experimental part is limited in different ways. The main critics regarding those are as follows:
1. The authors only use synthetic data that mostly comply with their assumptions, thereby confirming that their approach is valid. THey do not really challenge it.
2. There is no realistic data sets used (which is very difficult to obtain in incremental preference learning, but less in passive preference learning).
3. There is also no real comparisons with other methods intending to learn value functions from observed preferences. Given the large literature on preference learning and the fact that most learning papers require the comparisons with other baselines when possible, one could have hoped to have such comparisons.

Depsite that, the paper appears to be well-made with a relevant algorithmic contribution.